# Novel Aspects of the Immune Response Involved in the Peritoneal Damage in Chronic Kidney Disease Patients under Dialysis

**DOI:** 10.3390/ijms24065763

**Published:** 2023-03-17

**Authors:** Flavia Trionfetti, Vanessa Marchant, Guadalupe T. González-Mateo, Edyta Kawka, Laura Márquez-Expósito, Alberto Ortiz, Manuel López-Cabrera, Marta Ruiz-Ortega, Raffaele Strippoli

**Affiliations:** 1Department of Molecular Medicine, Sapienza University of Rome, Viale Regina Elena 324, 00161 Rome, Italy; 2Department of Epidemiology, Preclinical Research and Advanced Diagnostics, National Institute for Infectious Diseases L., Spallanzani, IRCCS, Via Portuense, 292, 00149 Rome, Italy; 3Cellular Biology in Renal Diseases Laboratory, IIS-Fundación Jiménez Díaz-Universidad Autónoma Madrid, 28040 Madrid, Spain; 4REDINREN/RICORS2040, 28029 Madrid, Spain; 5Cell-Cell Communication & Inflammation Unit, Centre for Molecular Biology “Severo Ochoa” (CSIC-UAM), 28049 Madrid, Spain; 6Premium Research, S.L., 19005 Guadalajara, Spain; 7Department of Pathophysiology, Poznan University of Medical Sciences, 10 Fredry St., 61-701 Poznan, Poland; 8IIS-Fundación Jiménez Díaz-Universidad Autónoma Madrid, 28040 Madrid, Spain

**Keywords:** peritoneal dialysis, inflammation, peritoneal fibrosis, immune system, kidney, COVID-19

## Abstract

Chronic kidney disease (CKD) incidence is growing worldwide, with a significant percentage of CKD patients reaching end-stage renal disease (ESRD) and requiring kidney replacement therapies (KRT). Peritoneal dialysis (PD) is a convenient KRT presenting benefices as home therapy. In PD patients, the peritoneum is chronically exposed to PD fluids containing supraphysiologic concentrations of glucose or other osmotic agents, leading to the activation of cellular and molecular processes of damage, including inflammation and fibrosis. Importantly, peritonitis episodes enhance peritoneum inflammation status and accelerate peritoneal injury. Here, we review the role of immune cells in the damage of the peritoneal membrane (PM) by repeated exposure to PD fluids during KRT as well as by bacterial or viral infections. We also discuss the anti-inflammatory properties of current clinical treatments of CKD patients in KRT and their potential effect on preserving PM integrity. Finally, given the current importance of coronavirus disease 2019 (COVID-19) disease, we also analyze here the implications of this disease in CKD and KRT.

## 1. Introduction

Chronic kidney disease (CKD) will become the fifth greatest global cause of death by 2040 and the second cause of death before the end of the century in those countries with longer life expectancy, as predicted by the Global Burden of Disease study and the Spanish Society of Nephrology [1,2,3]. Current therapies only retards CKD progression and many people present an increased risk of requiring kidney replacement therapy (KRT), of cardiovascular complications, and of all death causes [2,3]. KRT includes peritoneal dialysis (PD), hemodialysis, or renal transplantation. CKD has been identified as the most prevalent risk factor for the lethal coronavirus disease 2019 (COVID-19) [4]. Renal complications in severe acute respiratory syndrome (SARS) pandemic studies pointed out an active role of viral infection in the worsening prognosis of patients suffering acute kidney injury (AKI). Remarkably, the fatality rate of AKI patients with SARS viral infection is more than 90% [5].

Current PD therapy is focused on solute (toxin) removal and fluid balance. The repeated infusion of peritoneal dialysis fluids (PDFs) into the peritoneal cavity not only partially replaces renal function, but also induces an array of local and systemic untoward effects. During PD therapy, the peritoneal membrane (PM) is continuously exposed to supraphysiologic concentrations of glucose. In addition, the formation of glucose degradation products (GDPs) during heat sterilization of PDFs is believed to be key factors in the limited biocompatibility of PDFs. This chronic exposure to PDFs can induce several cellular and molecular responses in the PM, including sterile inflammation, loss of the mesothelial cell (MC) monolayer, submesothelial fibrosis, vasculopathy, and angiogenesis [6,7]. Moreover, peritoneal infections, mainly by episodes of acute bacterial peritonitis, can also occur during PD treatment and catheter maintenance [7]. All these events contribute to morphologic and functional PM transformation and ultimately technique failure often found in long-term PD patients [8]. On the other hand, the systemic effects of PDFs chronic exposure comprise a metabolic, inflammatory, and immune-modulatory burden. The impact of this to patient outcome has, however, not yet been properly delineated. Importantly, the newer PDFs, including biocompatible glucose-based solutions, icodextrin, taurine, and other recently proposed solutions based in alternative osmotic agents (stevia, xylitol, and L-carnitine), as well as the addition of potentially protective compounds, such as alanyl-glutamine supplementation, exert lower deleterious effects in the PM [9,10,11,12,13]. Despite PD therapy, CKD patients remain at high risk of poor outcomes. Most notably, cardiovascular events and markers of systemic inflammation are strongly associated with CKD patients [10,14]. The life expectancy of patients requiring KRT overall is inferior to that for most common cancers, including those originating in the breast, lung, and colon [15].

In this review, we discuss novel data about the role of immune cells and inflammation in the damage of the PM during PD therapy, both by infections and chronic exposure to PDFs, as well as potential therapeutic strategies targeting inflammation, including some aspects of current clinical treatments in CKD patients. Finally, given that CKD increases the risk of death in COVID-19 patients, and in turn, COVID-19 can complicate kidney damage, we also analyze here the implications of COVID-19 in KRT.

## 2. Cellular and Molecular Mechanisms Implicated in the Damage of the Peritoneum

Peritoneal damage depends on complex interactions between external stimuli, intrinsic properties of the PM, and subsequent activities of the local innate-adaptive immune system [16]. In PD patients, repeated exposure of the PM to PDFs can induce a local inflammatory response, that is mediated by the activation of different inflammatory pathways, such as the NLRP3 inflammasome and the nuclear factor-kappa B (NF-κB) pathways [17,18]. Several cell types are involved in this sterile inflammatory response, including the MCs themselves, neutrophils, mast cells, monocytes/macrophages (MØs), T lymphocytes, dendritic cells (DCs) and resident fibroblasts [16,19]. Furthermore, toll-like receptors (TLRs) mediate sterile inflammation by recognizing damage-associated molecular patterns (DAMPs) released by cellular stress [20]. In infectious conditions, there is an overproduction of cytokines that is mainly regulated by a complex network involving various immune cells [21]. PM is also potentially exposed to microorganisms’ infections due to its proximity to the intestine from where in pathological conditions microorganisms may reach the peritoneal cavity. Moreover, external factors may contribute to the entry of microorganisms into peritoneum space, such as medical actions of catheter positioning and maintenance, the practice of peritoneal dialysis, and abdominal surgery [22]. Postoperative peritonitis accounts up to 65% of abdominal infection observed in intensive care unit patients [23], so it is considered the most frequent form of intra-abdominal infection [24]. In PD patients, emergency treatment of this clinical complication by acute infections employs standardized protocols and has a high success rate, but patients experiencing peritonitis have increased technique failure and cardiovascular complication rates through ill-defined mechanisms [14,24].

### 2.1. Microorganisms Involved in Peritonitis Episodes during PD

Gram-positive bacteria of the skin play a major role in causing peritonitis episodes, with a minor role of Gram-negative bacteria, presumably originating from the enteric flora [25]. Viral infections of PM are scarcely reported in peritonitis cases due to the lack of standard diagnosis tests. Despite this, the suspect of viral infection occurs when peritonitis microbial culture results negative, an event occurring around 20% of the cases [26]. Literature reports cases of viral infection in peritonitis, such as coxackievirus B1 infection characterized by the presence of monocytosis in PD effluent [27]. It has also been demonstrated that viral haemorrhagic septicaemia virus (VHSV) can infect the peritoneal cavity, activating the expansion and differentiation of a particular class of resident IgM+ B cells [28]. Also, less studied are peritonitis caused by fungal infections. They account for between 1 and 15% of all PD-associated peritonitis episodes, constituting a serious complication for PD patients. Most of these episodes are caused by *Candida* species such as *Candida albicans* [29,30].

### 2.2. Receptors and Ligands Implicated in the Infection of the Peritoneum

Innate pattern recognition receptors (PRRs) are the sensors implicated in PM damage by infectious and endogenous causes. These include TLRs, a retinoic acid-inducible gene I (RIG-I)-like receptors, NOD-like receptors, and C-type lectin receptors. The intracellular signaling cascade activated by PRRs determines the expression of inflammatory mediators acting in the elimination of pathogens and infected cells [31]. PRRs are able to recognize molecules conserved among microbial species called pathogen-associated molecular patterns (PAMPs) as well as DAMPs [20,32]. Among PRRs, TLRs play a critical role in the innate immune response by specifically recognizing molecular patterns from a wide range of microorganisms, including bacteria, fungi, and viruses. TLRs are responsible for sensing invading pathogens outside of the cell and in intracellular endosomes and lysosomes [31]. Ten different TLRs in humans and twelve in mice have been identified so far. Each of them recognizes different molecular patterns of microorganisms and self-components. TLR2 and TLR5 recognize Gram-positive bacteria [33], both more singularly than cross-talking to better counteract bacterial infections [34]. TLR2 can recognize a large number of microbial molecules in part by hetero-dimerization with other TLRs, such as TLR1 and TLR6, or unrelated receptors, such as Dectin-1, CD36, and CD14. TLR5 recognizes flagellin, a flagellum component in many motile bacteria [35]. TLR4 was initially identified as the detector for lipopolysaccharide (LPS), inducing response against Gram-negative bacteria. A set of TLRs, comprising TLR3, TLR7, TLR8, and TLR9, act in the intracellular space to recognize nucleic acids derived from viruses and bacteria, as well as endogenous nucleic acids in pathogenic contexts [31]. TLR3 mediates the recognition of viral stimuli and is functionally expressed in peritoneal MCs [36]. TLR3 was demonstrated acting on fibrosis onset, in particular on matrix-remodeling proteins as it is correlated in time- and dose-dependent upregulation of matrix metalloproteinase (MMP) 9 and metalloproteinase inhibitor 1 (TIMP1) [37]. Several studies have demonstrated the active role of TLRs in modulating peritoneal fibrosis, particularly focusing on TLR2, critical for the recognition of an *S. epidermidis*-derived cell-free supernatant, which has been extensively utilized to model acute peritoneal inflammation [38,39]. Treatment with soluble Toll-like receptor 2, a TLR2 inhibitor, has also been shown to substantially reduce inflammation in an experimental in vivo model of *S. epidermidis* infection [20].

### 2.3. Role of Polymorphonucleate Neutrophils in Acute Infection and Early Innate Response

Pathogen-associated infection in the peritoneum first promotes a wave of polymorphonuclear neutrophils recruited by chemoattractants of bacterial origin or by chemokines, such as the C-X-C motif chemokine ligand (CXCL) 1 and CXCL8, produced mainly by MCs and omental fibroblasts. Neutrophils can use high endothelial venules present in anatomic structures called milky spots or fat-associated lymphoid clusters (FALCs) to enter the peritoneal cavity under the guidance of CXCL1 [40] (Figure 1). Neutrophil influx in the peritoneal cavity causes an initial inflammatory response driven by neutrophil-secreted proteases and reactive oxygen species (ROS). Secondly, once entered in peritoneum neutrophils undergo NETosis, which consists of the release of necrotic cell DNA forming a net of aggregated neutrophils able to trap and sequester microorganisms in FALCs, thus limiting their spreading [41]. Neutrophils also participated in the recruitment of mononuclear infiltrates through secreting IL6Rα, a shed form of interleukin (IL)-6 receptor. The local increase of IL-6Rα promotes an IL-6-mediated neutrophil clearance after mononuclear cell recruitment through a mechanism called trans-signaling [42,43]. Apoptotic neutrophils are phagocytized by MØs and to a lesser extent by the same MCs [44]. Necrotic neutrophils and NETs promote the infiltration of mature MØs recruited also by locally produced chemokines, such as CXCL8 and CCL2 [45]. Two other cytokines, IL-17A and interferon (IFN)-γ, also regulate neutrophil functions, including their influx in the peritoneum and clearance process [46].

### 2.4. Role of Polymorphonucleate Macrophages and Dendritic Cells in Acute Infection and Chronic Inflammation

Peritoneal tissue-resident cells include MØs and DCs, as components of the peritoneal immune system. They participate in the induction of inflammatory response, pathogen clearance, tissue repair, and antigen presentation [47,48].

MØs are plastic cells involved in different diseases, playing both essential protective and pathological roles [47,48]. MØs can present two different phenotypes, named M1 and M2, which exert different properties and functions. Briefly, M1 MØs are involved in amplifying the first phase of the inflammatory process creating a gradient of chemotactic cytokines, such as CXCL8, C-C motif chemokine ligand (CCL)-2, and CCL5, necessary for the recruitment of other leukocytes. This process is concomitant to cytokine-driven up-regulation of adhesion molecules expression (ICAM-1 and VCAM-1) on the surface of MCs, which facilitates leukocyte adhesion to MCs. Instead, M2 MØs actively counteract the inflammatory process through the production of soluble anti-inflammatory mediators, and the clearance of debris such as apoptotic or necrotic products, due to their scavenger activity [49].

MØs can be considered the major resident immune population in the PM where they play an essential for maintaining tissue homeostasis [50], being the predominant cell type found in dialysis effluent [51,52]. Under inflammatory or infection conditions, a depletion of resident peritoneal MØs populations has been reported [53], a phenomenon known as Macrophage Disappearance Reaction (MDR) [54]). In contrast, reducing the influx of monocytes derived MØs or depleting all MØs populations usually prevents injury in experimental models of peritoneal damage [55]. Primary peritoneal M2 MØs exhibited superior anti-inflammatory potential than immobilized cell lines. Interestingly, a high number of MØs, neutrophils, or a higher ratio of MØs/DCs can be associated with severe and recurrent episodes of peritonitis [51]. Moreover, peritoneal resident M2 MØs have been demonstrated to have an active role in attenuating the cytokine storm in severe acute infections [56].

A recent study showed that chronic exposure to PDFs alters resident MØs homeostatic phenotype, including the lost expression of anti-inflammatory and efferocytosis markers and enhanced inflammatory response [57]. Another study demonstrated the role of M1 instead of M2 MØs in peritoneal damage by depleting and transplanting different MØs populations into PDF-exposed mice [58]. In contrast, another study reported increased expression of the M2 MØs markers CD206, transforming growth factor-beta (TGF-β), Ym-1, and Arginase-1 in experimental PDF exposure, together with fibrosis markers, which were recovered by clodronate-liposome-mediated MØs depletion [59]. In a peritoneal damage model induced by sodium hypochlorite, inflammatory M2 MØs switch to CCL-17-expressing profibrotic phenotype to promote fibroblasts activation [60].

In patients on continuous ambulatory PD (CAPD), a decrease in peritoneal MØs function has been observed [61], together with alterations in MØs heterogeneity (i.e., different maturation and activation states), which have been associated with different PD outcomes [51]. Moreover, peritoneal MØs isolated from patients under CAPD during peritonitis episodes exhibit increased proinflammatory cytokines expressions, such as IL-1β and tumoral necrosis factor (TNF)-α, compared to macrophages from CAPD patients without peritonitis [62]. Additional studies showed that peritoneal CD206^+^CD163^+^ M2 MØs are present in PD effluents of PD patients with peritonitis and that these MØs contribute to peritoneal fibrosis by stimulating fibroblast cell growth and increasing the production of the soluble factor CCL-18 [63]. Thus, the role of MØs during PDF-induced peritoneal damage depends on the pathological context and MØs subtype. Therefore, therapeutic approaches targeting the overall MØs population must be evaluated as resident peritoneal MØs play a key role in peritoneal immunity.

### 2.5. Role of Th17 Lymphocytes and Its Effector Cytokine IL-17A in Acute Infection and Chronic Inflammation

Th17 cells are a type of differentiated T helper lymphocytes characterized by a specific secretome including cytokines of the IL-17 family, mainly IL-17A, its effector cytokine, and other Th17-specific cytokines, such as IL-22, IL-26, and CCL20 [64]. Th17 plays a key role in the response against pathogens but is also implicated in the pathogenesis of many inflammatory and autoimmune diseases, including psoriasis, rheumatoid arthritis, and multiple sclerosis [65]. Besides Th17 cells, IL-17A can be secreted by various leukocyte subpopulations, including neutrophils, Th1, and γδ-T lymphocytes and its expression correlated with the duration of the PD treatment and with the extent of peritoneal inflammation and fibrosis [46,66]. In peritoneal biopsies of PD patients, IL17-positive cells were identified as Th17 cells, γδ lymphocytes, mast cells, and neutrophils, and were positively correlated with peritoneal fibrosis [66].

Several experimental evidence shows the involvement of Th17/IL17A in acute peritoneal damage. In a murine model of intraperitoneal sepsis with abscess formation, an increase in the number of Th17 cells in the peritoneal cavity was observed, whereas the treatment with a neutralizing antibody against IL-17A prevented the abscesses formation [67]. Curiously, in peritonitis induced by *S. aureus* delivery and caecal ligation and puncture (CLP) in mice, γδ T lymphocytes, instead of Th17 cells, were found as the main source of IL-17A [68]. In the CLP model, intraperitoneal IL-17A and other cytokine levels were increased and IL-17A neutralization reduced proinflammatory cytokine levels and improved mice survival [69]. As commented before, IL-17A regulates several neutrophil functions [46]. In cultured human MCs, IL-17A has been shown to activate different inflammatory pathways, mainly the canonical NF-κB pathway and its downstream cytokines, such as granulocyte-colony stimulating factor (G-CSF) and growth-regulated alpha protein (GROα, also known as CXCL1), the latter involved in neutrophil infiltration into the peritoneum [70,71].

Besides its role in acute infections, Th17/IL-17A axis is also involved in chronic inflammation of the PM. In mice models of PDF exposure, a predominance of Th17 cells over Treg cells has been observed in the peritoneum [72], as well as activation of the transcription factors involved in Th17 differentiation retinoic acid-related orphan receptor (RORγt) and Signal transducer and activator of transcription 3 (STAT3) [66]. Moreover, exposure to PDF induces local production of Th17-related cytokines (IL-17A and IL-6), together with a correlation between peritoneal IL-17A protein levels and membrane thickness [66]. In addition, studies conducted in gene mice deficient for CD69, a leucocyte protein that modulates Th17 cell differentiation, the chronic exposure to PDFs highly increased the Th17 response and IL-17A production leading to exacerbation of the inflammatory and fibroproliferative response [73,74]. In these PDF-exposure models, the treatment with neutralizing anti-IL-17A antibodies improved peritoneal damage [66,74]. Other study of PDF-exposure in uremic mice showed that conventional PDF treatment increased CD4+/IL-17+ cells in the peritoneal cavity [75], while a biocompatible low-GDP bicarbonate/lactate-based PDF increased peritoneal recruitment of M1 macrophages and decreased number of CD4+/IL-17+ cells, which were associated with better preservation of PM integrity [75]. These experimental studies support the detrimental effect of an enhanced Th17 response and IL-17A expression in the PDF-exposed peritoneum, whereas Th17 modulation and IL-17 targeting could be an effective therapeutic approach for peritoneal damage.

In peritoneal effluents from patients on PD using different conventional PDF, IL-17A has been found to increase, together with other cytokines such as IL-6, IL-1β, TNF-α, and TNF-like weak inducer of apoptosis (TWEAK) [76,77,78]. In peritoneal biopsies from PD patients, IL-17A-positive cells were found in the submesothelial zone overlapping with inflammatory and fibrosis areas. Interestingly, IL-17A levels in the effluents were significantly higher in long-term PD patients after 3 years of dialysis [66]. On the other hand, during peritonitis episodes IL-17A peritoneal effluents levels may reach a huge increase [79], showing the involvement of IL-17A in chronic and acute peritoneal damage.

Altogether, these pieces of evidence support the contribution of Th17 cells and IL-17A-producing cells to peritoneal inflammation associated with dialysis.

### 2.6. Other Cells Implicated in the Peritoneal Response to Infections: Mastocytes and Natural Killer Cells

The role of mastocytes, another resident leukocyte population, in peritoneal damage is controversial. A study performed in a mast cell-deficient rat model demonstrated that mast cells increase omental thickness and adhesion formation favoring leukocyte recruitment [80]. Studies performed in PD patients showed an increased number of mastocytes in samples from different inflammatory and fibrotic peritoneal diseases, including PD and encapsulating peritoneal sclerosis (EPS) [81]. Studies in peritonitis demonstrated that natural killer (NK) cells have a role in neutrophil apoptosis and subsequent inflammation resolution [82,83]. For example, kidney dysfunction with induction of fibrosis and CKD progression correlates with activation of tubulointerstitial human CD56^bright^ NK [84]. However, studies on humans are limited. NK cells in combination with IL-2 can cause peritoneal fibrosis in patients with malignancies [85].

## 3. Resident Peritoneal Cells: Activation of Proinflammatory and Immunomodulatory Signals

Peritoneum is composed of a monolayer of MCs with an epithelial-like cobblestone shape covering a continuum of the peritoneal cavity. Respect to its huge extension and its unique localization in the abdominal cavity, the peritoneum is a favorable site for the encounter of antigens and for the subsequent generation of the immune response. Resident MCs have a mesodermal origin, and their differentiation is controlled by the transcription factor Wilms’ tumor 1 (WT1) [86,87]. MCs can undergo a process called mesothelial to mesenchymal Transition (MMT) in which they acquire mesenchymal features such as mobility and tissue invasion. The intrinsic plasticity of these cells is linked to their ability to acquire mesenchymal-like features in response to a variety of proinflammatory and profibrotic stimuli. Almost all the pro-inflammatory factors described in the previous section may promote the induction of MMT in MCs. This dedifferentiation process permits MCs to acquire morphological and functional features making these cells indistinguishable from myofibroblasts of other origin. High throughput experiments have correlated the expression of profibrotic and proinflammatory cytokines, such as TGFβ1, vascular endothelial growth factor A (VEGFA), and IL-6 to the induction of MMT [20,88,89,90]. In fact, activated MCs are major producers of those cytokines, whose concentrations are elevated especially during peritonitis, and have been associated with ultrafiltration decline and protein loss [25,91]. The secretion of these cytokines impacts fibrosis, angiogenesis, and the inflammatory response.

The first phases of the peritoneal immune response not only involve resident peritoneal leucocytes [92,93]. MCs composing parietal and visceral peritoneum play an active role in the response to infections first directly responding to extracellular mediators released by microorganisms [94]. Human peritoneal MCs (HPMCs) express a specific subset of TLRs composed of TLR1, TLR2, and TLR3, with a moderate or scarce level of TLR4 and TLR10 [33]. A comparison of peritoneal MCs from murine and human origin responsiveness revealed important differences between the two species. The major difference consists of the unresponsiveness of HPMCs to TLR4 ligands, unlike murine peritoneal MCs (MPMCs). However, TLR4 is expressed by human MØ stably residing in the PM, and their response may contribute to inflammation leading to fibrosis [95,96]. Instead, HPMCs and MPMCs showed similarities in TLR1/2, TLR2/6 and TLR5 ligand responsiveness [95,96].

MCs are the main constituent of FALCs, whose frequency and size increase in the peritoneum of patients undergoing PD [97,98]. Besides MCs, FALCs are composed of MØs, and B1 cells. B1 cells have the potential to produce natural antibodies that provide a first protection against bacterial infections [99]. The chemokine CCL19 produced by structural components of FALCs is extremely relevant for monocyte recruitment during inflammation, activating a crosstalk that promotes T cell dependent-B cell immune response [100]. Thus, FALCs play a main role both in the regulation of polymorphonuclear (PMN) and mononuclear cell recruitment in the first phase of inflammation, as well as in the subsequent induction of the adaptive immunity.

Neutrophils, for example, use high endothelial venules present in FALCs to enter the peritoneal cavity, guided by the chemotactic signal of CXCL1 [40]. The production of CXCL1 as well as other chemoattractant CXCL8 by the peritoneal stroma is enhanced by inflammatory cytokines such as IL-1β and to a lesser extent, TNFα [101]. Stimulation of MCs with IL-1β or LPS, for example, also induces the production of a number of cytokines and chemokines including IL-6, TNFα, CCL2, CCL3, favoring mononuclear cell recruitment and activation [102].

Peritoneal mesothelium-derived chemokines have been recently proposed as potential therapeutic targets. In this sense, CXCL1 induced in MCs in response to IL-17 exposure displays angiogenic activity and can subsequently control the vascular remodeling in dialyzed peritoneum [103]. The observation of PD patients exhibiting high CXCL1 expression and subsequent density of microvessels opens to new intervening prospective acting to preserve long-term PM integrity and function.

Another important source of extracellular mediators engaged in peritoneal immunity are resident fibroblasts, embedded in the submesothelial stroma. Their role is mainly associated with the development of peritoneal fibrosis induced by chronic inflammation [104]. However, human peritoneal fibroblasts (HPFBs) upon stimulation with macrophage or T lymphocyte-derived proinflammatory cytokines, such as IL-1β, TNFα and IFN-γ, secrete large quantities of interleukins and chemokines (CCL2, IL-8/CXCL8, IL-6, CXCL1, CCL5) [101,105,106,107,108]. Therefore, HBFBs control trans-peritoneal chemotactic gradients during peritonitis, crucial when MCs are damaged and exfoliated.

## 4. Cellular Senescence in the Peritoneum

Cellular senescence is a cellular program initiated by various forms of stress that further may jeopardize the integrity of the genome. It occurs after a period of vigorous proliferation and is characterized by irreversible growth arrest, altered morphology, and different patterns of gene expression. Growth arrest in senescent cells is related to the upregulation of at least two major tumor suppressor pathways: the p53/p21 and p16INK4a/pRB [109]. The known triggers of senescence include telomere dysfunction, oncogene activation, reactive oxygen species, and epigenomic damage [110]. As a response to injury, such as shortened telomeres which are recognized as double-strand breaks (DSBs), the DNA damage response (DDR) is initiated leading the cell to replicative senescence [111].

Senescent cells remain viable and metabolically active. However, changes in gene expression lead to different secretion of numerous cytokines, chemokines, growth factors, proteases, and extracellular matrix proteins, a feature known as the senescence-associated secretory phenotype (SASP) [112]. The induction of SASP on the transcriptional level is regulated mainly by C/EBP (CCAT/Enhancer Binding Protein) and NF-κB transcription factors. C/EBPβ has been shown to regulate many SASP components, such as IL-1β, IL-8, IL-6 or CXCL1/GROα [113]. In addition, the SASP is positively regulated by the NF-κB that controls the transcription of many genes encoding proinflammatory cytokines. This regulation applies to IL-6 and IL-8 which are viewed as the most prominent mediators of the SASP [112]. The secreted SASP modifies the tissue microenvironment and is able to promote the senescence phenotype to neighboring cells in a paracrine manner [114,115]. It is well established, that peritoneal MCs when entering replicative senescence in vitro, secrete large amounts of proangiogenic (VEGF, CXCL1), proinflammatory (IL-6, IL-8, MCP-1/CCL2) and the profibrotic factors TGF-β and connective tissue growth factor (CTGF/CCN2), precisely known as the SASP components (Figure 2) [116]. Different preclinical studies have demonstrated that these components of the SASP can significantly contribute to adverse peritoneal remodeling during PD by promoting MMT and angiogenesis [46,103,117,118].

Furthermore, an intriguing feature of senescent HPMCs was described. In those cells, both markers typical for mesenchymal and epithelial phenotype were observed, which usually indicates a partial MMT process [119]. Thus, there is a possible phenotypical connection between senescence and MMT and inflammatory factors that can induce either of the processes.

There is increasing evidence suggesting that senescent cells contribute to both organismal ageing and age-related pathologies, and accumulation of senescent cells may promote both degenerative and hyper-proliferative disorders, whose incidence rises exponentially with age [109]. The occurrence of cellular senescence in the peritoneum in vivo is poorly documented. It was found that impaired viability and function of HPMCs exposed to glucose-containing PDF was predominantly related to the presence of GDPs and, further to the presence of glucose [120]. However, more recent studies have shown that glucose can lead to the destruction of the integrity of HPMCs, triggering apoptosis and inhibiting cell regeneration [121]. Detailed examination of the mesothelium of rats treated intraperitoneally with high loads of glucose revealed the presence of enlarged senescence-like cells [122]. Cells expressing senescence-associated-β-galactosidase (SA-β-Gal), a well-defined marker of senescence, were also found in freshly explanted human omenta [123], in MCs imprints from mice exposed to PDFs [124] or in MCs derived from PD effluents [125]. A more recent study conducted on omental arterioles of patients treated with PDFs revealed that a high concentration of glucose degradation products in the peritoneum lead to upregulation of p16 and disruption of lamin A/C in nuclear envelope [126].

Taken together, these data indicate that by producing several extracellular mediators within the SASP, the senescent cell population contributes to the immunomodulation of the peritoneal environment during PD.

## 5. Potential Anti-Inflammatory Treatments in the Preservation of the PM Integrity

Many preclinical data have already demonstrated the beneficial effects of current clinical treatments of CKD patients in the preservation of PM integrity. Therefore, we will briefly review here whether the current clinical treatments of CKD patients exert anti-inflammatory effects, as well as the data supporting the effect of some of these drugs in the preservation of the PM.

### 5.1. Current Clinical Treatments in CKD Patients That Exert Anti-Inflammatory Actions

Although many preclinical studies targeting inflammation have described kidney protective effects, most of the promising anti-inflammatory drugs undergoing clinical testing for CKD [127,128,129], have failed or clinical development has stopped and to date, no anti-inflammatory drug is in clinical use [129]. However, drugs in clinical use display anti-inflammatory properties that at least in some cases are the direct result of their mechanism of action. Interestingly, most of these observations have been made in diabetic kidney disease (DKD), i.e., in an environment characterized by high glucose concentrations [129] and, thus, may be relevant for PM injury in the setting of high concentrations of glucose or GDPs. Renin-angiotensin-aldosterone system (RAAS) inhibitors, including angiotensin-converting enzyme (ACE) inhibitors (ACEIs) and angiotensin receptor blockers (ARBs), are prescribed as antihypertensive drugs and for kidney protection in diabetic and non-diabetic kidney disease [130,131,132]. Moreover, they preserve residual kidney function in PD patients [133]. Many preclinical studies have demonstrated anti-inflammatory effects of both drug families, through inhibition of Angiotensin II-induced deleterious effects, including the activation of key inflammatory-related mechanisms, including oxidative stress, NF-κB, janus kinase (JAK)/signal transducer and activator of transcription (STAT) and TLRs pathways [132]. More recently, clinical trials have shown kidney and cardiovascular protection by sodium-glucose cotransporter-2 (SGLT2) inhibitors in both diabetic and non-diabetic kidney disease [134,135]. Currently, guidelines support the use of SGLT2 inhibitors on top of RAAS blockade in diabetic and non-diabetic kidney disease [132,136]. The nonsteroidal mineralocorticoid receptor antagonist (MRA) finerenone also improved kidney and cardiovascular outcomes in DKD, even in patients already on RAS blockade and SGLT2 inhibitors [137,138]. Again, clinical guidelines support their use in DKD [132]. Additionally, aldosterone synthase inhibitors are under clinical development [139]. Both SGLT2 inhibitors and MRAs have decreased kidney inflammation in vivo [140,141,142].

The endothelin receptor antagonists (ERA) reduced inflammatory macrophage infiltration in preclinical studies [143]. While the ERA Atrasentan improved kidney outcomes in DKD, it failed to improve cardiovascular outcomes and clinical development for this indication was stopped [144,145,146]. However, the dual ARB-ERA Sparsentan has shown promising results for proteinuric kidney disease [147] and the combination of RAS blockade, SGLT2 inhibition, and ERA is undergoing clinical trials in DKD since the protection from heart failure offered by SGLT2 inhibition is expected to make ERA safer and the kidney protective effect may be additive [148].

Complement represents a key part of the innate immune response and has been involved in nephropathies ranging from acute kidney injury to DKD [149,150,151,152] Several complement-targeting therapies are in clinical use for kidney disease, including undergoing clinical trials for an expanding number of nephropathies. Eculizumab and Ravulizumab, targeting C5, are used for hemolytic uremic syndrome while Avacopan, a C5a receptor inhibitor, is used for antineutrophil cytoplasmic antibody (ANCA)-associated vasculitis [153,154]. Targeting cytokines has been widely investigated in autoimmune diseases, but only a few studies have addressed human CKD [129], among them, IL-17A has emerged as an interesting candidate [155].

### 5.2. Anti-Inflammatory Actions of Current Clinical Treatments for CKD in the Preservation of the PM Integrity

In preclinical studies, administration of RAAS inhibitors (such as Aliskiren, Valsartan, Enalapril, and Lisinopril) by different routes is able to reduce peritoneal thickening and improve peritoneal function in PDF exposure models in rats [156,157,158,159]. Components of the RAAS are constitutively expressed within peritoneal MCs and are upregulated during acute inflammation and chronic exposure to PDFs [160]. The high glucose concentration, low pH, and the formation of GDPs in PDFs have all been associated with peritoneal RAAS modulation. Besides, activation of the RAAS contributes to MMT of MCs, resulting in progressive fibrosis of the PM. This process also leads to increased VEGF production, promoting peritoneal angiogenesis. Functionally, these changes reduce the ultrafiltration capacity of the PM, leading to PD technique failure [160].

Findings on experimental animal models suggest that part of the renoprotective effects of SGLT2 inhibition may be related to anti-inflammatory actions in the kidney [161]. Thus, its potential anti-inflammatory effect might also be taken advantage of at the peritoneal level. Remarkably, SGLT2i may have multiple benefits for CKD patients undergoing PD in terms of glycemic, atherosclerotic process, and peritoneal deterioration control [162]. In this regard, SGLT2 is expressed in MCs and in skeletal muscle. Dapagliflozin, an SGLT2 inhibitor, significantly reduced effluent TGF-β concentrations, peritoneal thickening, and fibrosis, as well as microvessel density, resulting in improved ultrafiltration in a mouse PD model. In vitro, Dapagliflozin reduced monocyte chemoattractant protein (MCP)-1/CCL2 release under high-glucose conditions in human and murine peritoneal MCs [163].

Endothelin (ET)-1 and its receptors are expressed in the PM and result in upregulated during PDFs exposure. In a PD model, administration of an ERA, either Bosentan or Macitentan, markedly attenuated PD-induced MMT, fibrosis, angiogenesis, and peritoneal functional decline [164].

Some MRAs have been tested to prevent peritoneal inflammation and fibrosis in animal models. Spironolactone was shown to reduce both processes in a rat model of PD on intraperitoneal injection of dialysates and LPS, and another rat model based on mechanical scraping of the peritoneum. Furthermore, peritoneal function assessed by the peritoneal equilibration test was significantly improved by spironolactone [165,166]. Eplerenone was administered to rats with intraperitoneal exposure to LPS, and the results were compared with those obtained in animals treated with PDFs or PDFs + LPS. Interestingly, ultrafiltration and transperitoneal osmotic diffusion were significantly impaired by LPS and restored by eplerenone. Increased value of the mass transfer area coefficients for creatinine values was also recovered by Eplerenone [167].

### 5.3. Novel Anti-Inflammatory Treatments in Experimental PD

In the last years, novel therapeutic agents have been assayed in preclinical models of peritoneal damage. These approaches have been shown to reduce peritoneal damage and membrane thickening by targeting different components and molecular pathways involved in the inflammatory response (Figure 3).

#### 5.3.1. Inhibition of Classical Proinflammatory Cytokines and Chemokines

NF-κB signaling pathway is one of the most activated inflammatory pathways in response to damage to the peritoneum. Very recent studies have shown that the treatment with the NF-κB inhibitor Parthenolide improved peritoneal damage induced by PDF exposure [168,169]. This improvement included decreased peritoneal levels of inflammatory cytokines, such as IL-6, TNF-α, and MCP-1, and suppression of TGF-β/SMAD signaling activation. Another pathway involved in peritoneal inflammation and injury induced by PDFs is the JAK2/STAT3 signaling [170]. In cultured HPMCs and MPMCs, the JAK2/STAT3 inhibitor WP1066 prevented MMT induced by IL-6 and DOK silencing, respectively [171,172]. PDF exposure also led to TNF-α signaling activation, including increased levels of TNF-α and significant upregulation of TNF Receptor 1 (TNFR1) on the mesothelial cell surface [18]. In transgenic mice expressing human TNFR1, the peritoneal inflammation and fibrosis induced by PDF exposure was attenuated using a monoclonal antibody (H398) that selectively blocks human TNFR1 [173]. In a model of chlorhexidine gluconate (CG)-induced damage, CCL8 was the most upregulated chemokine in the peritoneum and the functional blockade of CCL8 using a CCR8 inhibitor (R243) prevented PM thickness and macrophage infiltration into the peritoneum [174].

#### 5.3.2. Glycogen Synthase Kinase 3 Beta Inhibition

Glycogen synthase kinase 3 beta (GSK3β) has been associated with inflammation and fibrogenesis in different tissues [175,176]. In PDF-exposed mice, a peritoneal hyperactivity of GSK3β (characterized by increased GSK3β expression and reduced inhibitory phosphorylation at serine 9) has been found [177]. Accordingly, the therapeutic targeting of GSK3β by Salvianolic Acid A improved peritoneal fibrosis together with attenuated expression of the inflammatory cytokines IL-1 β, TNF-α, and MCP-1, prevented activation of NF-κB signaling pathway, reinforced the nuclear factor-erythroid 2-related factor 2 (NRF2) antioxidant response, and diminished oxidative injury [177]. Moreover, lithium chloride (LiCl), a known GSK3β inhibitor, has been shown to be cytoprotective in peritoneal damage. LiCl-supplemented PDF promoted morphological preservation of MCs and the submesothelial zone in chronically PDF-exposed mice by suppressing αB-crystallin [178].

#### 5.3.3. Cyclooxygenase-2/ Prostaglandin-E2 Pathway Blockade

The cyclooxygenase-2 (COX-2)/prostaglandin-E2 (PGE2) pathway is a key player in PDF-induced PM inflammation [179]. Celecoxib, an inhibitor of COX-2 activity, has been shown to improve PM ultrafiltration capacity and prevent peritoneal fibrosis, inflammation, and angiogenesis in murine models and PD patients [179,180,181]. On the other hand, the blockade of PGE2 receptor 4 (EP4) with ONO-AE3-208, an EP4 receptor antagonist, diminished the activation of NLRP3 inflammasome and phosphorylation of p65 NF-κB subunit together with decreased peritoneal fibrosis and improved peritoneal dysfunction in a PD model in rats [182].

#### 5.3.4. Targeting Autophagy

Autophagy is a vital mechanism to maintain cell homeostasis under physiological conditions, but it is also involved in the physiopathology of several diseases [183]. In long-term PD, autophagy promotes fibrosis and apoptosis in the peritoneum [184]. In two different murine models of PDFs and CG damage, autophagy was found highly activated [185]. In these models, autophagy inhibition with 3-methyladenine (3-MA) successfully prevented peritoneal damage, noted by downregulation of TGF-β/SMAD signaling and the EMT transcription factors Slug and Snail, repressed activation of epidermal growth factor receptor (EGFR)/extracellular signal-regulated protein kinases ½ (ERK1/2) signaling pathway, decreased STAT3/NF-κB-mediated inflammatory response and macrophage infiltration, and prevented β-catenin-mediated peritoneal angiogenesis [185]. In contrast, another study showed that treatment with trehalose, an autophagy inducer, has beneficial effects on CG-induced peritoneal fibrosis, including decreased levels of α-SMA, Collagen I α1, and Snail, but with no effect on SMAD or mitogen-activated protein kinase (MAPK) signaling pathways [186]. In accordance with this, other researchers reported that mice treated with high glucose-PDF showed decreased expression levels of the autophagy-related proteins Beclin-1 and microtubule-associated proteins 1A/1B light chain 3B (LC3-II) and upregulated expression of p62; while the treatment with 1,25(OH)2D3, the active form of vitamin D, attenuated peritoneal damage by enhancing autophagy [187,188]. In this line, paricalcitol, an analog of vitamin D and a specific activator of vitamin D receptors (VDR), has been shown to fully recover the loss of ultrafiltration capacity induced by PDF exposure in rats, together with a decrease in ECM thickening and reduced angiogenesis [189]. In mice, paricalcitol reduced PDF-induced peritoneal by modulating Th17 and Treg response [190]. Rapamycin, another known autophagy inducer, has been shown to exert protective effects of PDF exposure models, these effects include a reduced PM thickness, decreased peritoneal fibrosis, improved PM transport function [191], reduced angiogenesis and lymphangiogenesis [192], decreased MMT and endothelial-to-mesenchymal transition [192,193] and improved lipid metabolism [194].

#### 5.3.5. Targeting Mitochondrial Dysfunction and Oxidative Stress

Several studies have reported that oxidative stress and mitochondrial dysfunction play an essential role in PDF-associated peritoneal damage [195,196]. In human peritoneal MCs, high glucose PDF induced excessive ROS production and lipid peroxidation with oxidative DNA damage [197]. In mice, PDF exposure downregulated the adenine monophosphate (AMP) activated protein kinase (AMPK)- peroxisome proliferator activated receptor-gamma coactivator-1 alpha (PGC-1α) signaling pathway, including decreased expression of p-AMPK, PGC-1α, nuclear factor-erythroid 2-related factor (NRF)-1, NRF-2, and the mitochondrial transcription factor A (TFAM), as well as reduced levels of mitochondrial DNA and damaged mitochondrial structure in the peritoneum [198]. In this model, treatment with metformin, an AMPK agonist, improved all these parameters and inhibited apoptosis of peritoneal MCs with the consequent improvement of peritoneal fibrosis [198]. Moreover, mitochonic acid-5 (MA-5), an indole-3-acetic acid derivative, has been shown to improve mitochondrial dysfunction and peritoneal damage in a model of peritoneal fibrosis induced by CG [199]. In this study, MA-5 was able to decrease α-SMA-positive myofibroblasts, TGF-β-positive cells, F4/80-positive macrophages, MCP1-positive cells, and 4-hydroxynonenal (4-HNE)-positive cells was considerably decreased in addition to increased ATP synthase subunit alpha (ATP5A1)-positive and uncoupling protein-2 (UCP2)-positive cells [199]. Another study showed that molecular hydrogen, which selectively scavenges cytotoxic ROS and acts as an antioxidant, administrated in hydrogen-rich PDF successfully inhibited the peritoneal fibrosis progress induced by high glucose PDF by removing intracellular ROS and inhibiting the activation of the phosphatase and tensin homologue deleted on chromosome 10 (PTEN)/protein kinase B (AKT)/ (mammalian target of rapamycin) mTOR signaling pathway [200]. On the other hand, protein kinase C (PKC) is known to induce oxidative stress by induction of ROS production mediated by NADPH oxidases [201]. In a mice model of chronic exposure to high-glucose PDF, the blockade of PKCα by intraperitoneal administration of the conventional PKC inhibitor Go6976 or genetic deficiency prevented peritoneal damage, including reduction of proinflammatory, profibrotic, and proangiogenic mediators such as MCP-1, TGF-β, and VEGF, respectively [202].

#### 5.3.6. Modulation of Epigenetic Mechanisms

Increasing evidence has demonstrated the involvement of epigenetic mechanisms regulating gene expression in several diseases, including CKD and peritoneal damage [203,204]. These epigenetic mechanisms include DNA methylation, posttranslational histone modifications, and non-coding RNAs [203].

Regarding DNA methylation, only one study reported aberrant gene methylation in peritoneal damage. In this study, increased protein levels of DNA methyltransferase 1 (DNMT1) were observed in the peritoneum of rats treated with CG, together with hypermethylation of Ras GTPase activating-like protein 1 (RASAL1) gene and decreased RASAL1 protein levels [205]. Notably, the treatment with the demethylating agent 5′-azacytidine restored RASAL1 methylation and expression, together with the improvement of peritoneal fibrosis [205].

In addition, alterations in different histone-modifying enzymes have also been reported in peritoneal damage [204]. In a model of peritoneal fibrosis induced by methylglyoxal (MGO) in mice, the H3K9 methyltransferase G9a has been found to increase, whereas G9a blockade using the BIX01294 inhibitor decreased submesothelial zone thickness, fibrosis, and infiltration of monocytes, TGF-β levels, and PM function [206]. Similarly, the inhibition of the H3K4 methyltransferase SET7/9 using Sinefungin also improved MGO-induced peritoneal fibrosis in mice [207]. Moreover, in mice models of CG- and PDF-exposure, the blockade of the H3K27 methyltransferase enhancer of zeste homolog 2 (EZH2) using 3-deazaneplanocin A (3-DZNeP) reduced peritoneal fibrosis by preventing the activation of several profibrotic signaling pathways, including TGF-β/SMAD3, Notch1, and epidermal growth factor (EGF), together with decreased inflammation noted by less STAT3 and NF-κB phosphorylation and reduced lymphocyte and macrophage infiltration [208]. In vitro, hyperacetylation of H3 histone was observed in HPMCs during high glucose-induced MMT, and treatment with the histone acetyltransferase (HAT) inhibitor C646 reversed MMT of HPMCs via blocking TGF-β1/SMAD3 signaling [209].

However, no in vivo experiments have been reported so far using this inhibitor. Conversely, several studies demonstrated the role of histone deacetylases (HDACs) in the control of peritoneal fibrosis. In cultured PD effluent-derived MCs, treatment with MS-275, an HDAC1-3 inhibitor, restored an epithelial signature after PDF-induced MMT. This included downregulation of the mesenchymal markers MMP2, collagen type 1 alpha 1 (COL1A1), plasminogen activator inhibitor (PAI)-1, and TGFβ1 and its receptor type 1 (TGFβRI); upregulation of the epithelial markers E-cadherin and occludin, and the loss of invasive features [210]. Moreover, treatment with MS-275 as well as HDAC1 genetic silencing induced the expression of WT1, a transcription factor controlling MC differentiation [211].

*In vivo*, the delivery of suberoylanilide hydroxamic acid (SAHA), an HDAC inhibitor, prevented the progression of CG-induced fibrosis in mice, noted by reduced submesothelial thickness and type III collagen accumulation, and less FSP1- and α-SMA-positive cells in the peritoneum [212]. Additionally, the treatment with Tubastatin A, a highly selective HDAC6 inhibitor, in PDF-exposed mice, prevented submesothelial zone thickness and decreased expression of collagen type I, α-SMA, and TGF-β1, and inhibited the phosphorylation of SMAD3, EGFR, STAT3, and p65 NF-κB [213]. In a model of CG-induced peritoneal fibrosis, Tubastatin A also prevented damage, including reduced fibrosis and inhibition of M2 macrophages polarization by suppressing the activation of TGF-β/SMAD3, PI3K/AKT, STAT3, and STAT6 pathways [214].

Micro RNAs (miRNAs) are the most studied non-coding RNAs. Different miRNAs have been found in PD effluents and have been associated with peritoneal function and damage parameters [215]. Most of them have been studied *in vitro*, whereas few studies have been performed on in vivo models. In fibroblastic MCs from PD patients, miR-769-5p has been demonstrated to directly target TGFBRI, SMAD2/3, and PAI-1 expression and to promote the reacquisition of epithelial-like features [211]. Among miRNAs studied in in vivo models, miR-21-5p blockade using an inhibitor anti-miRNA-21-LNA reduced PM thickness, decreased the expression of the fibrosis markers Col1a1, Fsp1, and Fn1, and increased the expression of the anti-inflammatory response transcription factor PPAR-α [216]. Moreover, treatment with an inhibitor of miR-296-3p improved peritoneal fibrosis and angiogenesis in a high glucose PDF-exposure model in mice [172]. On the other hand, the expression of miR-302c, a miRNA that negatively correlated with CTGF and MMT markers in MCs from PD effluents of patients, was found to be downregulated in the peritoneum of PDF-exposed mice and the transfection of a lentivirus containing mmu-miR-302c protected the PM from PDF-induced fibrosis improving CTGF, Collagen type I, and α-SMA levels [217].

## 6. COVID-19 in CKD Patients under KRT and in Mesothelium

At the end of 2019, a novel coronavirus (CoV) designated as SARS-CoV-2 emerged in the city of Wuhan, China, and caused an outbreak of unusual viral pneumonia. It is now well assessed that SARS-CoV-2 first infects epithelial cells of the upper respiratory tract (nasal passages and throat) and especially lungs (bronchi and alveoli), where alveolar type I and type II cells are believed to mediate the first encounter with the virus; the subsequent infection of alveolar macrophages is determinant in mediating the amplification of the inflammatory and immune responses [218]. SARS-CoV-2 pathogenesis, characterized by clinical phenotypes spanning from asymptomatic infection to mild disease with symptoms related to airways tract implication, severe pneumonia, acute respiratory distress syndrome (ARDS), and multiple organ failure, has been largely studied during the last three years [219]. Several studies pointed out the role of viral secondary targets implied in the worsening of the pathology. The virus can penetrate in the blood circulation resulting in secondary organ infection. A molecular investigation on COVID-19 autopsies demonstrated the presence of SARS-CoV-2 secondary infection in several organs, detecting high viral positivity in the nasopharynx (90.4%) followed by bilateral lungs (87.30%), peritoneal fluid (80%), pancreas (72.72%), bilateral kidneys (68.42%), liver (65%) and even in the brain (47.2%) [220]. Literature reports the kidney as one of the most probable secondary organs for SARS-CoV-2 infection [221,222,223,224].

Mechanistically, SARS-CoV-2 entrance in the host cell is mediated by ACE2 as a main receptor, although ACE2-independent entry has been demonstrated [218]. Molecular analysis in human tissues exhibited higher expression of ACE2 in the kidney than in the lung [225,226,227]. One of the causes of kidney injury in COVID-19 patients can be ascribed to a specific immunological effector response, such as specific T lymphocytes, induced by the virus or by the interaction between viral antigens with specific antibodies [228,229]. SARS-CoV-2 implication in the complication of kidney pathologies was also demonstrated by cohort studies [230,231,232,233]. To overcome SARS-CoV-2 effects in kidney deterioration, hemodialysis has been proposed [234]. Importantly, CKD, after old age, is the factor that most increases the risk of death in COVID-19 patients [4].

COVID-19 also affects patients on KRT. Some studies have demonstrated susceptibility to infection in patients undergoing PD, although the reported cases are less frequent than in hemodialyzed patients [235]. However, PD could represent a more practical and safer KRT method for CKD patients, considering a home-safer option during the pandemic crisis [236]. In this sense, a recent cohort population study of PD patients pointed out that this KRT does not increase the risk of infection for SARS-CoV-2. Most of the infected patients studied successfully developed specific viral antibodies and get recovered from COVID-19, maintaining PD treatment [237]. Remarkably, SARS-CoV-2 infection positively correlated with PD failure, suggesting a possible effect of the virus on the PM [238]. Although it is conceivable that the PM is a secondary infected organ in SARS-CoV-2 pathogenesis, there is no in vivo direct evidence so far of MCs infection by the virus. Despite this, MCs have been found to express SARS-CoV-2 specific receptors/co-receptors ACE2, transmembrane serine protease 2 (TMPRSS2), A disintegrin and metalloproteinase 17 (ADAM17), and neuropilin 1 (NRP1) [239].

M2 MØs has been previously discussed for their immunomodulatory/anti-inflammatory activity during peritoneum infections. Accordingly, treatment with extracellular vesicles (EVs) generated in response to SARS-CoV-2 infection reduced the expression of inflammatory cytokines, such as TNF-α and IL-6, released in vitro and in vivo, attenuating oxidative stress and multiple organs (lung, liver, spleen, and kidney) damage in endotoxin-induced cytokine storms [56]. Moreover, M2 peritoneal MØs were also demonstrated to release high levels of ACE2 in EVs and this could represent a decoy strategy to prevent SARS-CoV-2 infection [56]. Similarly to MØs, pleura MCs have been demonstrated to produce anti-inflammatory cytokines in vitro in response to SARS-CoV-2 viral infection [239]. In fact, IL-10 and TNFRs were shown to represent a predominant cytokine response in SARS-CoV-2 MCs infected cells, over proinflammatory cytokines such as IL-1β, TNFα, and IL-6. This overproduction of anti-inflammatory cytokines could have systemic relevance due to the extension of the pleura surface (2000 cm^2^ in an average adult male) as a homeostatic mechanism of dampening the inflammatory response during infection.

The availability of SARS-CoV2 vaccines brought new hope and changed the perspective of the COVID-19 pandemic. COVID-19 vaccination was demonstrated to reduce severe complications related to SARS-CoV-2 infection [240]. However, since the generation of a specific immune response is needed for a positive response to vaccinations, the efficacy of vaccinations may be diminished in patients with CKD, due to a possible impairment of the immune system in these patients [241]. With respect to kidney transplant recipients, it was reported that most of them could not generate a protective humoral immunity after standard protocols of SARS-CoV-2 vaccination due to the fact that they require treatments with immunosuppressive drugs [242]. However, seroconversion is not an accurate measure of response, since some patients who were not able to produce antibodies still had a detectable vaccine-specific T-cell response, which might be sufficient to prevent severe COVID-19 [243,244]. Importantly, infection or a third dose of mRNA vaccine was demonstrated to elicit neutralizing antibody responses against SARS-CoV-2 in kidney transplant recipients [245].

Thus, the availability of SARS-CoV2 vaccines changed the natural history of the COVID-19 pandemic strongly reducing severe life-threatening complications. However, in immunocompromised patients such as transplanted patients, vaccination may fail to induce a protective immune response. In this case, personalized vaccination regimens and/or alternative pharmacological treatment are urgently needed.

## 7. Conclusions

Immune cells are responsible for PM damage during repeated exposure to PD fluids not only during bacterial or viral infections but also in sterile conditions. Different already used drugs in CKD with anti-inflammatory properties have been proposed here as interesting options to serve the dual purpose as clinical treatments for CKD and to help preserving PM integrity during PD treatments in these patients.

The irruption of the SARS-CoV-2 pandemic constituted a new challenge in the treatment of CKD patients. CKD increases the risk of death in COVID-19 patients, but also SARS-CoV-2 viral infection can complicate kidney damage by different mechanisms. In this condition, the home-based PD practice could represent a more practical and a safer KRT method for CKD patients.

## Figures and Tables

**Figure 1 ijms-24-05763-f001:**
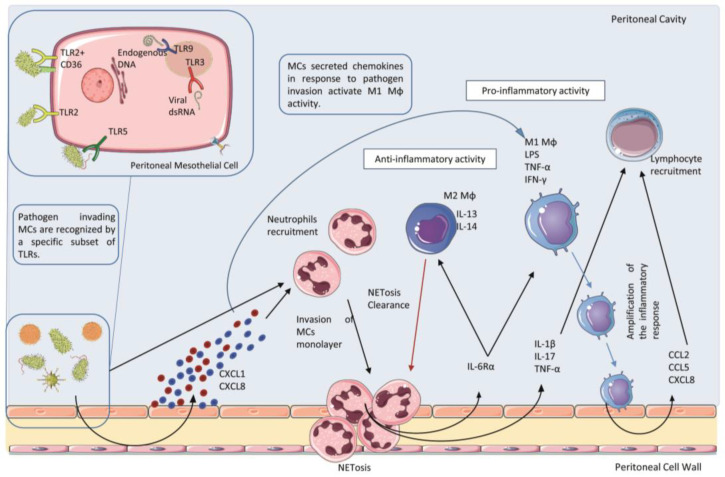
Acute infection and early innate response in the peritoneum. Mesothelial cells (MCs) can sense invading microorganisms through a specific subset of Toll-Like Receptors (TLRs). Pathogen infection stimulates MCs to produce chemokines, such as CXCL1 and CLXL8, which promote the recruitment of a first wave of neutrophils entering the peritoneal cavity and undergo to the process of NETosis. Cytokines and chemokines released both by MCs and Neutrophils during this first wave of the inflammatory process induce the recruitment of mononuclear phagocyte. Macrophages (MØs) recruited in this area display different phenotype subtypes. M1 subtype shows pro-inflammatory and cytotoxic properties, whereas the M2 subtype presents an anti-inflammatory activity. Moreover, M2 MØs play a key role in the clearance of neutrophils debris due to scavenger activity. To activate the adaptative immune response, M1 MØs secrete CCL2, CCL5 and CXCL8 which acts as a chemoattractant for lymphocyte recruitment in the peritoneal cavity.

**Figure 2 ijms-24-05763-f002:**
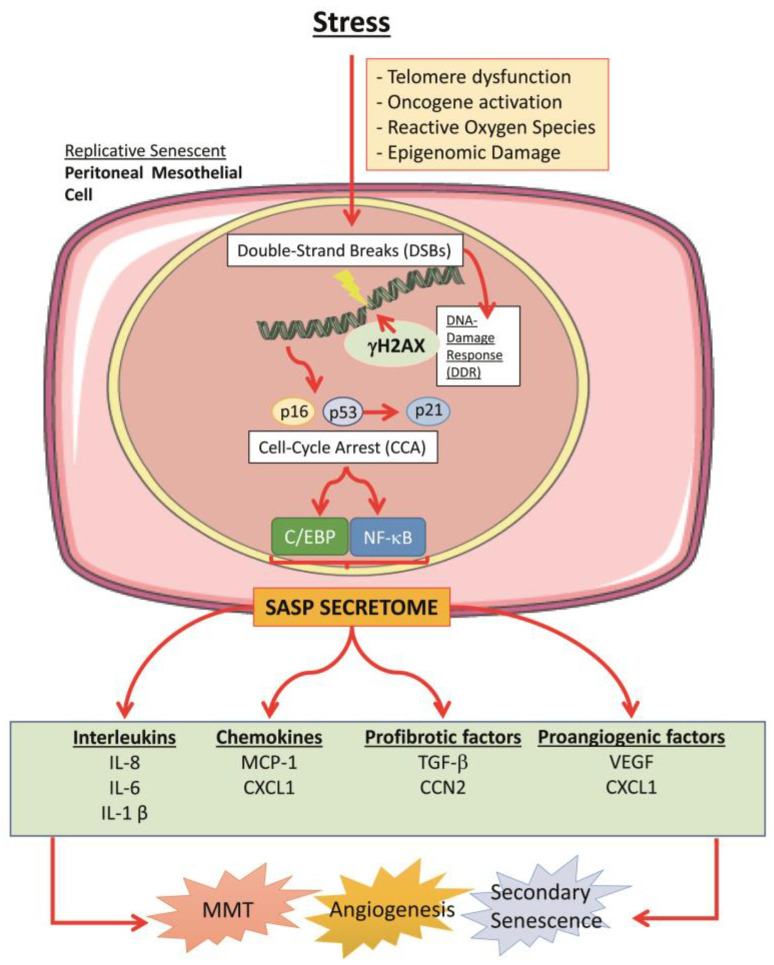
Cellular senescence is triggered by different kinds of stimuli, some resulting in Double-Strand breaks (DSBs). This activates the DNA-Damage Response (DDR), in which γH2AX may repair the DNA. However, prolonged damage leads to Cell-Cycle Arrest (CCA) due to p16, p53, and p21 upregulation. Then, the cell gene expression is reprogramed, starting to release the senescence-associated secretory phenotype (SASP), mainly regulated by C/EBP (CCAT/Enhancer Binding Protein) and NF-κB (nuclear factor-κB), and enriched with numerous factors such as interleukins, chemokines, profibrotic and proangiogenic mediators. These metabolic changes can contribute to several pathological processes in the peritoneal cavity, such as Mesothelial-to-Mesenchymal transition (MMT), angiogenesis, and secondary senescence, or spreading cellular senescence to neighboring cells.

**Figure 3 ijms-24-05763-f003:**
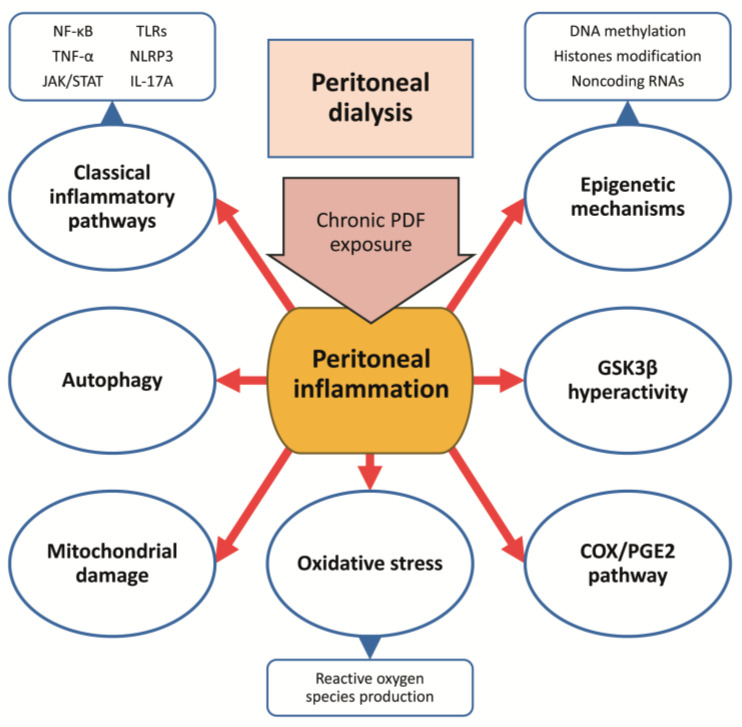
Cellular pathways and mechanisms involved in the inflammatory response to chronic PDF exposure during PD.

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
