# Peer review of "Novel Aspects of the Immune Response Involved in the Peritoneal Damage in Chronic Kidney Disease Patients under Dialysis"

_ijms, 2023, doi:10.3390/ijms24065763_

Round 1

Reviewer 1 Report

This review is comprehensive and well written. it covers all the major discussion points for the role of the immune response in pathogenesis of chronic kidney disease. This review covers not only the pathology but also therapeutic aspects of the topics and will be interesting to the users to read. 

Author Response

Thank you very much for your words and your work reviewing our manuscript.

Reviewer 2 Report

Comments to the authors

Lines 39 to 41. How can authors mention CKD as the second cause of death before the end of the century? Is there any scientific evidence?

It would be good to combine parts 4.1. Current clinical treatments in CKD patients that exert anti-inflammatory actions and 4.2. Anti-inflammatory activities of current clinical treatments for CKD in the preservation of the PM integrity, since authors mention the same story.

In the conclusion part, out of two paragraphs, the authors overstate SARS-CoV-2 infection in the final paragraph. What message do authors want to convey from that?

Author Response

Comments to the authors

Lines 39 to 41. How can authors mention CKD as the second cause of death before the end of the century? Is there any scientific evidence?

Thank you very much for your question. We have based this sentence on studies and publications based on the Global Burden of Disease (GBD) study. The first reference of the manuscript stated that CKD will become the second cause of death before the end of the century, according to the Spain GBD 2016 report (Med Clin (Barc) 2018; 151: 171–190) and the Spanish Society of Nephrology (SENEFRO) commentary to this report (Nefrologia 2019; 39: 29–34). According to these studies, CKD was the 8th cause of death in Spain in 2016 and, among the top ten causes of death, it was the fastest growing from 2006 to 2016. Based on this growth, the growth up to the year 2100 can be predicted. Now we have included these data in the introduction

It would be good to combine parts 4.1. Current clinical treatments in CKD patients that exert anti-inflammatory actions and 4.2. Anti-inflammatory activities of current clinical treatments for CKD in the preservation of the PM integrity, since authors mention the same story.

Thank you very much for your recommendation. We have decided to include the section 4.2 as a sub-section of 4.1 (now 4.1.2). Because the first part (4.1) describes the current drugs that exert anti-inflammatory effects exclusively in CKD, and the second part (now 4.1.2) is focused on the anti-inflammatory effect of these drugs on peritoneum.

In the conclusion part, out of two paragraphs, the authors overstate SARS-CoV-2 infection in the final paragraph. What message do authors want to convey from that?

We have changed the second and last paragraph of our conclusion to highlight the challenge that the COVID-19 pandemic has represented in the treatment of CKD and PD patients.

Reviewer 3 Report

In this interesting review by Flavia Trionfetti et al., they describe the involvement of immune response cells in peritoneal dialysis in patients with CKD or CRRT and provide information on the complications presented by patients with renal disease and SARS-COV2 viral infection.

Authors are encouraged to consider the following comments for the improvement of their manuscript:

1.     Overall, the wording of the manuscript is adequate, but authors are advised to revise their revision because moderate changes in English are required. For example, the sentence “Persons with CKD present increased risk of progressing to require kidney replacement therapy (KRT), of all-cause and cardiovascular death and of acute kidney injury (AKI)” in lines 41 and 42 is not understood.

2.     In the sentence "Several cell types are involved in this sterile inflammatory response, including MCs themselves, neutrophils, dendritic cells, mast cells, monocytes/macrophages (MØ), T lymphocytes, dendritic cells (DCs) and resident fibroblasts" on lines 85-87, the cell type "dendritic cell" is repeated twice.

3.     Authors are encouraged to add a description to Figure 1.

4.     The authors are suggested to discuss further the impact of COVID-19 vaccines on chronic kidney disease and kidney transplantation patients.

Author Response

In this interesting review by Flavia Trionfetti et al., they describe the involvement of immune response cells in peritoneal dialysis in patients with CKD or CRRT and provide information on the complications presented by patients with renal disease and SARS-COV2 viral infection.

Authors are encouraged to consider the following comments for the improvement of their manuscript:

  1. Overall, the wording of the manuscript is adequate, but authors are advised to revise their revision because moderate changes in English are required. For example, the sentence “Persons with CKD present increased risk of progressing to require kidney replacement therapy (KRT), of all-cause and cardiovascular death and of acute kidney injury (AKI)” in lines 41 and 42 is not understood.

Thank you very much for your suggestion. We have changed this sentence for the following: “Current therapies only retards CKD progression and many people present increased risk of requiring kidney replacement therapy (KRT), cardiovascular complications and death of all causes”.

  1. In the sentence "Several cell types are involved in this sterile inflammatory response, including MCs themselves, neutrophils, dendritic cells, mast cells, monocytes/macrophages (MØ), T lymphocytes, dendritic cells (DCs) and resident fibroblasts" on lines 85-87, the cell type "dendritic cell" is repeated twice.

 Thank you very much for your suggestion. We have corrected that in the text.

  1. Authors are encouraged to add a description to Figure 1.

 As the reviewer suggested, we have added a description for the Figure 1.

  1. The authors are suggested to discuss further the impact of COVID-19 vaccines on chronic kidney disease and kidney transplantation patients.

Thank you so much for your suggestion. We have added a paragraph discussing the impact of COVID-19 vaccines on CKD and kidney transplanted patients.